# Perturbation-response analysis of in silico metabolic dynamics revealed hard-coded responsiveness in the cofactors and network sparsity

**Yusuke Himeoka[1]\*, Chikara Furusawa[1,2]**

[1]Universal Biology Institute, University of Tokyo, Tokyo, Japan; [2]Center for Biosystems Dynamics Research, RIKEN, Osaka, Japan

## eLife Assessment

This **valuable** study uses dynamic metabolic models to compare perturbation responses in a bacterial system, analyzing whether they return to their steady state or amplify beyond the initial perturbation. The evidence supporting the emergent properties of perturbed metabolic systems to network topology and sensitivity to specific metabolites is **solid**.

**\*For correspondence:**
yhimeoka@g.ecc.u-tokyo.ac.jp

**Competing interest:** The authors declare that no competing interests exist.

**Abstract** Homeostasis is a fundamental characteristic of living systems. Unlike rigidity, homeostasis necessitates that systems respond flexibly to diverse environments. Understanding the dynamics of biochemical systems when subjected to perturbations is essential for the development of a quantitative theory of homeostasis. In this study, we analyze the response of bacterial metabolism to externally imposed perturbations using kinetic models of *Escherichia coli*'s central carbon metabolism in nonlinear regimes. We found that three distinct kinetic models consistently display strong responses to perturbations; in the strong responses, minor initial discrepancies in metabolite concentrations from steady-state values amplify over time, resulting in significant deviations. This pronounced responsiveness is a characteristic feature of metabolic dynamics, especially since such strong responses are seldom seen in toy models of the metabolic network. Subsequent numerical studies show that adenyl cofactors consistently influence the responsiveness of the metabolic systems across models. Additionally, we examine the impact of network structure on metabolic dynamics, demonstrating that as the metabolic network becomes denser, the perturbation response diminishes—a trend observed commonly in the models. To confirm the significance of cofactors and network structure, we constructed a simplified metabolic network model underscoring their importance. By identifying the structural determinants of responsiveness, our findings offer implications for bacterial physiology, the evolution of metabolic networks, and the design principles for robust artificial metabolism in synthetic biology and bioengineering.

## Introduction

For bacterial cells steadily growing under fixed, substrate-rich culture conditions, several quantitative laws pertaining to the physiology of growing bacteria have been identified. These include the linear association between growth rate and the ribosomal fraction in total proteomes (*Schaechter et al., 1958*; *Scott et al., 2010*), the linear relationship linking substrate uptake to growth rate (*Pirt, 1965*),

and the correlations between transcriptional responses to various sublethal stresses (*Kaneko et al., 2015*).

In recent years, the dynamic aspects of bacterial physiology have garnered significant attention. A wealth of intriguing phenomena under such conditions has been gradually revealed. For instance, cells can 'remember' the duration of starvation and the nature of environmental changes—such as the rate at which nutrients deplete—and this can dramatically alter the distribution of lag times (*Kaplan et al., 2021*; *Himeoka and Kaneko, 2017*; *Levin-Reisman et al., 2010*; *Himeoka et al., 2022a*). Both the proteome and metabolome of starved cells display characteristically slow dynamics (*Radzikowski et al., 2016*). The metabolites used for storage play a pivotal role in fluctuating environments (*Sekar et al., 2020*). Further, there is a trade-off between the growth rate under nutrient-rich conditions and other physiological parameters, namely, the lag time upon nutrient shifts (*Basan et al., 2020*) and the death rate under starved conditions (*Biselli et al., 2020*).

To achieve a comprehensive understanding of physiological responses, the development of quantitative theories that elucidate the dynamics of cellular physiology is indispensable. Metabolism, often a primary trigger for physiological transitions, stands at the forefront of efforts to forge a unified perspective on the dynamics of bacterial physiology. To this end, extensive research has been undertaken to understand the dynamic responses of metabolic systems using mathematical frameworks; the metabolic control analysis (*Heinrich and Rapoport, 1974*), the biochemical systems theory, especially S-systems (*Savageau, 1988*), and the linear stability analysis of mass-balance kinetic models (*Chakrabarti et al., 2013*; *Lee et al., 2014*). The exploration of the study on the stability and responsiveness of the metabolic system is limited only to the linear region except S-systems. However, the S-system approach omits the mass-balance. The mass-balance condition, or, in other words, the stoichiometric constraints, is known to have a strong impact on the dynamics of chemical reaction systems (*Feinberg, 2019*). For the development of theoretical foundations of metabolic dynamics, it is essential to explore the dynamic responses of mass-balancing metabolism beyond the linear regime.

In this article, we examine the mass-balancing kinetic models of *Escherichia coli*'s central carbon metabolism and ask how the metabolic state responds to perturbations with which the linear approximation is no longer valid. We study three independently proposed kinetic models (*Chassagnole et al., 2002*; *Khodayari et al., 2014*; *Boecker et al., 2021*) and explore the features shared among the models. We demonstrate that the three models exhibit strong responses to perturbations on the metabolic states. The effects of these perturbations are magnified, leading to significant deviations in metabolite concentrations from their steady-state values, depending on which and how metabolite concentrations are perturbed. The observed responsiveness is a hallmark of the realistic metabolic network as the toy metabolic models lack such a strong response. Computational analysis revealed that cofactors, such as ATP and ADP, play a crucial role in the strong response to perturbations. We also discovered the importance of the sparse structure of the metabolic network by adding virtual reactions to it. To validate our findings, we developed a simple, minimal model of metabolic reactions, and confirmed that cofactor dynamics and network sparsity are indeed the key ingredients of the strong responses.

## Results
### Model

In the present study, we investigate the dynamic response of metabolic states to perturbations in metabolite concentrations using kinetic models of *E. coli* central carbon metabolism. In kinetic modeling approaches, the temporal evolution of metabolites' concentrations is modeled using ordinary differential equations to capture behaviors in out-of-steady-state metabolism.

We employ three kinetic models of *E. coli*'s central carbon metabolism, emphasizing the common features exhibited by the models. A limitation of kinetic models is their specificity; unlike constraint-based modeling, kinetic modeling necessitates extensive biochemical information, such as the function form of the reaction rate equation and the parameter values for each reaction. By examining a single kinetic model, conclusions may apply only to that specific model. Therefore, our focus lies on the shared features of the three models to circumvent the limitations of individual model specificity. We also believe there is potential for the results obtained commonly among the models to apply to real biological systems as well.

The models under consideration are those proposed by *Chassagnole et al., 2002*, *Khodayari et al., 2014*, and *Boecker et al., 2021*. While all three models incorporate the glycolytic pathway, only the models by Khodayari et al. and Chassagnole et al. feature the pentose phosphate (PP) pathway. Notably, the Chassagnole model excludes the tricarboxylic acid cycle. A graphical summary of the metabolic modules in these models is presented in *Figure 1A*.

The Boecker and Chassagnole models utilize glucose as the sole carbon source, whereas the Khodayari model can assimilate other carbon sources, such as fructose, formate, and acetate depending on the differences between the extracellular and intracellular concentrations. The Boecker model explicitly models biomass formation, considering several metabolites as biomass precursors that are consumed with a fixed stoichiometry to produce biomass. Conversely, the Khodayari and Chassagnole models do not incorporate the biomass formation reaction. Instead, biomass precursor metabolites are consumed independently, implying that the stoichiometry of biomass formation varies depending on the concentrations. The dilution effect due to the volume growth is modeled in both the Boecker and Chassagnole models. In the Boecker model, the dilution rate corresponds proportionally to the biomass formation reaction rate, while it remains constant in the Chassagnole model. The Khodayari model omits the dilution effect. In addition to the metabolic reaction implemented in each model, the specific forms of the reaction rate equations also differ among the models.

Within these models, the dynamics of gene regulation mediated by transcription factors are not modeled; that is, the total concentration of each metabolic enzyme is constant while the substrate-level regulations are incorporated into the models (the lists of substrate-level regulations are provided in *Supplementary file 1a–c*; for a detailed description, see 'Materials and methods').

## Perturbation-response simulation

In this section, we present the results of a series of simulations, which we call perturbation-response simulations. These simulations highlight the unique responses of the metabolic models to perturbations and shed light on the dynamic behavior of metabolic systems. It may be useful for the reader to first describe the perturbation-response simulation procedure. The procedure is consistently applied across all three models used in our study and can be summarized as follows:

1. Compute the growing steady-state attractor.
2. Generate $N_{ini}$ initial points by perturbing the metabolite concentrations from the attractor.
3. Simulate the model dynamics starting from each initial point.

### 1. Computing the attractor

First, we identify the steady-state attractor where the production/consumption of each metabolite is balanced. For the Khodayari and Boecker models, the attractor corresponds to the steady state studied in their original papers. For the Chassagnole model, we numerically determined the steady-state attractor (details provided in 'Materials and methods').

### 2. Generating the initial points by perturbation

Once the steady-state attractor has been computed, we establish a set of initial points for subsequent computations by perturbing the metabolite concentrations. This perturbation is a proxy for stochastic fluctuations within cells. We postulate that the perturbation source in the variability in protein concentrations is due to the inherent randomness of transcription and translation processes. Also, the cell division would be another source of perturbations of metabolites' concentrations while the cell division is not explicitly considered in the study. Based on a stochastic model of transcription and translation with a biologically relevant setup, the relative size of the concentration fluctuations is estimated to be several tens of percent (for a detailed derivation, see 'Materials and methods'). Consequently, the perturbed concentration of chemical $n$ is $x_n = r_n x_n^{st}$, where $r_n$ is a uniformly distributed random number with a maximum strength of 40%, that is, ranging from 0.6 to 1.4. The perturbation strength is beyond the linear region that the relaxation dynamics is approximated by the linearized model around the steady state. The choice of uniform distribution is not essential for the following result. Also, the characteristic features of the response studied later remain consistent even when different perturbation strengths are employed (see *Figure 1—figure supplement 2*).

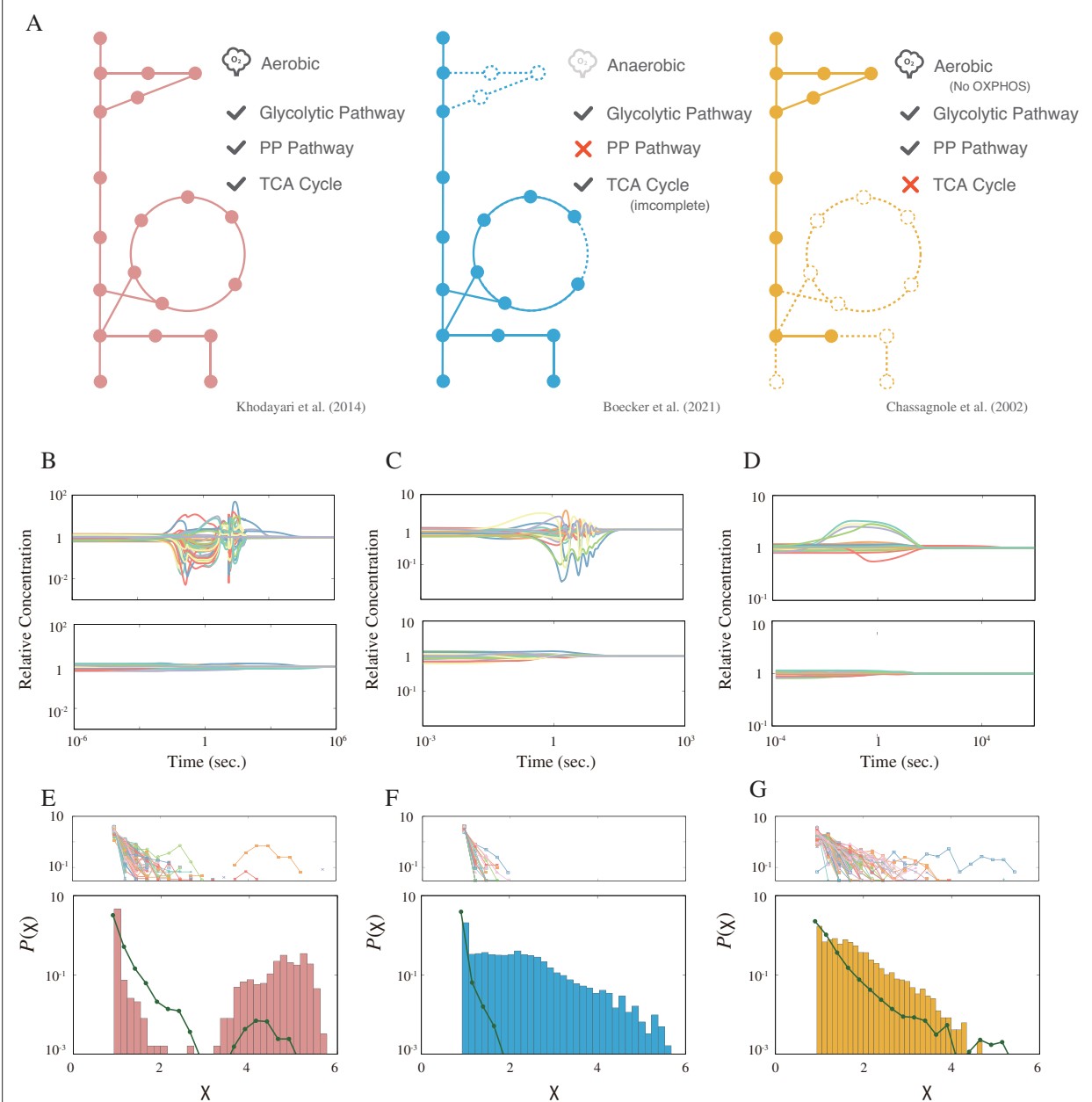

**Figure 1.** Models, dynamics, and distribution of the response coefficient. (**A**) A graphical summary of the three models used in this study. The solid circles represent the metabolites, and the lines represent the metabolic reactions. The dashed circles and lines are the subsystems not implemented in the model. (**B–D**) Example time courses for each model. The top and bottom panels are examples of the strong and weak response, respectively. (**E–G**) The response coefficient distribution $P(\chi)$ of each model (bottom). The average response coefficient distribution of the random catalytic reaction network (RCRN) model is overlaid (green line). The response coefficient distribution of each instance of the RCRN model is shown on the top panel. The model schemes, time courses, and distributions are aligned in each column, that is, the leftmost network of (**A**), (**B**), and (**E**) are from the model developed by **Khodayari et al., 2014**.

The online version of this article includes the following figure supplement(s) for figure 1:

**Figure supplement 1.** Scatter plot of the response coefficient.

**Figure supplement 2.** The response coefficient distribution of each model obtained by the perturbation-response simulation with different perturbation strengths (20%, 40%, and 60% relative perturbation).

**Figure supplement 3.** The time course of each model with the strongly responding metabolites being highlighted.

**Figure supplement 4.** The response coefficient distribution of 215 realizations of the PYK model by the MASSpy framework (**Himeoka et al., 2022a**) is overlaied (each model has the same network structure, kinetic rate law, but the parameters are different).

## 3.Simulating the dynamics

We compute the model dynamics using the $N_{\text{ini}}$ points generated by the perturbation as initial points to explore the metabolic response to the perturbations. As far as we have tried, all initial points have returned to the original growing steady state, and thus, we hereafter study the relaxation behavior.

## Strong responses in the kinetic models

After executing the perturbation-response simulations for the models, we observed distinct relaxation dynamics in each. In *Figure 1B–D*, the top and bottom panels display typical examples of dynamics from each model that either respond strongly or weakly to perturbations. In the weakly responding dynamics (bottom panel), the effects of the perturbations diminish almost monotonically. Conversely, in the dynamics that respond strongly (top panel), the initial displacements of concentrations are amplified over time, leading some metabolites' concentrations to overshoot or undershoot (significant concentration changes in metabolites are highlighted in *Figure 1—figure supplement 3*).

For more in-depth analysis, we introduce a measure of the responsiveness to the perturbation that we call the response coefficient $\chi$, defined as

$$\chi = \max_t \frac{\| \ln \boldsymbol{x}(t) - \ln \boldsymbol{x}^{\text{st}} \|}{\| \ln \boldsymbol{x}(0) - \ln \boldsymbol{x}^{\text{st}} \|},$$

where $\| \cdot \|$ is the Euclidean norm. The response coefficient distributions for each model are plotted in the bottom panels of *Figure 1E–G*. All three models have the peak at $\chi = 1$ (note that this is the minimum value of $\chi$). These peaks correspond to the trajectories where the Euclidean distance between the state and the attractor decreases monotonically with time (i.e., $\max_t \| \ln \boldsymbol{x}(t) - \ln \boldsymbol{x}^{\text{st}} \| = \| \ln \boldsymbol{x}(0) - \ln \boldsymbol{x}^{\text{st}} \|$). In addition, the distribution of the Khodayari model exhibits another peak in a larger $\chi$ region while the other two models have plateaus adjacent to the primary peak. We have computed the response coefficient by linearizing each model. As shown in *Figure 1—figure supplement 1*, the responsiveness of the original model cannot be described by the linearized model.

For comparison purposes, we conducted perturbation-response simulations for the random catalytic reaction networks (RCRN) model, a toy model of the metabolic network (*Furusawa and Kaneko, 2003*; *Furusawa and Kaneko, 2012*). In one instance of the RCRN model, the 'metabolites' are interconnected through a random network, and every metabolite is considered a catalyst as well as a reactant. We generated 128 instances of the RCRN model, each having the same number of metabolites and the same distribution of the reaction rate constant. For instance, to draw a comparison with the Khodayari model, we generated the model instances with 778 metabolites and 3112 reactions, where the reaction rate constants are randomly chosen from that of the Khodayari model (detailed construction of the RCRN model is provided in 'Materials and methods'). The top panel of *Figure 1E–G* presents the response coefficient distribution for each RCRN model instance, represented by different lines. Additionally, the 'average' response coefficient distribution, which is the mean of the distributions plotted in the top panel, is overlaid (green dotted line). As a fact, the emergence of the additional peak and plateaus, indicative of strong responsiveness, are unique traits of the realistic metabolic reaction network model and are seldom seen in toy representations of the metabolic network.

## Key metabolites on the responsiveness

What factors contribute to the strong responses of the metabolic state? In this section, we investigate which metabolites play pivotal roles in inducing these strong responses in the model. To understand the role of each metabolite's dynamics, we conducted perturbation-response simulations using model equations derived by fixing each metabolite's concentration to its steady-state value, that is, set $x_{met} = x_{met}^{\text{st}}$ and $dx_{met}/dt = 0$ for the concentration of the metabolite *met* whose concentration is fixed. The impact of holding each metabolite's concentration constant is quantified by the change in the average response coefficient:

$$\rho_x^+ = \frac{\langle \chi_x \rangle - \langle \chi_{\text{ori}} \rangle}{\langle \chi_{\text{ori}} \rangle}, \quad (\langle \chi_x \rangle \geq \langle \chi_{\text{ori}} \rangle) \tag{1}$$

$$\rho_x^- = \frac{\langle \chi_{\text{ori}} \rangle - \langle \chi_x \rangle}{\langle \chi_{\text{ori}} \rangle}, \quad (\langle \chi_x \rangle \leq \langle \chi_{\text{ori}} \rangle) \tag{2}$$

where $x$ denotes the metabolites' IDs and $\chi_x$ represents the response coefficient of the model with the concentration of the metabolite $x$ fixed to constant. Note that for each metabolite in each model, either the $\rho_x^+$ or $\rho_x^-$ is defined, depending on whether the modified model (with the metabolite $x$ as a held constant) exhibits an average response coefficient larger (for $\rho_x^+$) or smaller (for $\rho_x^-$) than the original model.

*Figure 2* displays $\rho_x^{\pm}$ values for representative metabolites (comprehensive results for all metabolites can be found in *Figure 2—figure supplement 1*). While most metabolites have a bilateral effect on the average response coefficient depending on the original model (or exert a small unilateral effect), ATP and ADP (we refer to them collectively as AXPs) consistently demonstrate a marked impact in reducing the average response coefficient across models. This observed significance of AXPs in metabolic dynamics aligns with previous findings (*Himeoka and Mitarai, 2022b*). The common importance of AXPs is shown also when we quantify the impact of their dynamics on the responsiveness by utilizing the Sobol' total sensitivity index (*Figure 2—figure supplement 2*; *Homma and Saltelli, 1996*).

Interestingly, halting the dynamics of several metabolites can amplify the model's response coefficient. Across the models, metabolites that augment the response coefficient when their concentrations are fixed tend to be allosteric regulators associated with negative regulation. Freezing these metabolites' concentrations effectively removes the negative feedback loop present in the metabolic systems. Typically, negative feedback stabilizes system behavior. In the present context, it assists the system in returning to its original state after perturbations. Hence, fixing these metabolites' concentrations allows the system to respond more vigorously to perturbations compared to the original model. We checked the effect of feedback itself on the responsiveness rather than the concentration dynamics by weakening the strength of substrate-level regulation of OAA, FUM, AKG in the Boecker model (*Figure 2—figure supplement 4*). It was shown that weakening the substrate-level regulation of CSICD by AKG and MDH by OAA increases the responsiveness. On the other hand, weakening the strength of substrate-level regulation of neither FRD by OAA nor MDH by FUM alters the responsiveness, and in addition there is no substrate-level regulation by SUC. The increase in responsiveness by fixing the concentrations of OAA and AKG would be explained by substrate-level regulation of the corresponding metabolites. However, substrate-level regulation is not sufficient to describe the effect of fixing the concentration of FUM and SUC. Higher order effects may be required.

Pyruvate is an exception to the above description, which is not the regulator in the models. The strong influence of pyruvate on the increase in response coefficients is attributed to the phosphotransferase system (PTS). The external glucose is taken up by converting phosphoenolpyruvate into pyruvate through the PTS. This reaction elevates pyruvate concentration, subsequently decelerating glucose uptake. By holding pyruvate concentration constant, this negative feedback effect is negated, enhancing the autocatalytic nature of the reaction system formed by the PTS and glycolysis ($\mathrm{GLC_{ex}} + \mathrm{PEP} \rightarrow \cdots \rightarrow 2\mathrm{PEP}$).

## Role of the sparsity of the networks

The key role of AXPs in changing the kinetic models' responsiveness is shown in the previous section. Nevertheless, this does not imply that all kinetic models of chemical reaction systems with cofactors such as AXPs always exhibit a strong response to perturbations. The strong responses demonstrated by the three models (as seen in *Figure 1B–D*) are absent in abstract toy models of metabolism, which consist of random reaction networks incorporating such cofactor chemicals *Kondo and Kaneko, 2011*. This discrepancy prompts us to investigate other potential determinants causing metabolic models to display strong responses shown in the three models.

A noteworthy attribute of metabolic networks is their inherent sparsity. While cofactors are linked to numerous reactions, the backbone networks—networks excluding cofactors—are distinctly sparse. As an example, the glycolytic pathway predominantly mirrors a linear sequence of reactions, resulting in many metabolites within this pathway participating in merely two reactions.

Does network structure play a role in the dynamics of metabolic systems? For this question, previous research in the opposite direction—reaction dynamics on dense networks—provides useful insights. Several studies use the random networks as an abstract model of metabolic networks (*Furusawa and Kaneko, 2003*; *Kaneko et al., 2015*; *Awazu and Kaneko, 2009*; *Himeoka et al., 2022a*). In such types of models, the effect of the initial perturbations on metabolite concentrations typically

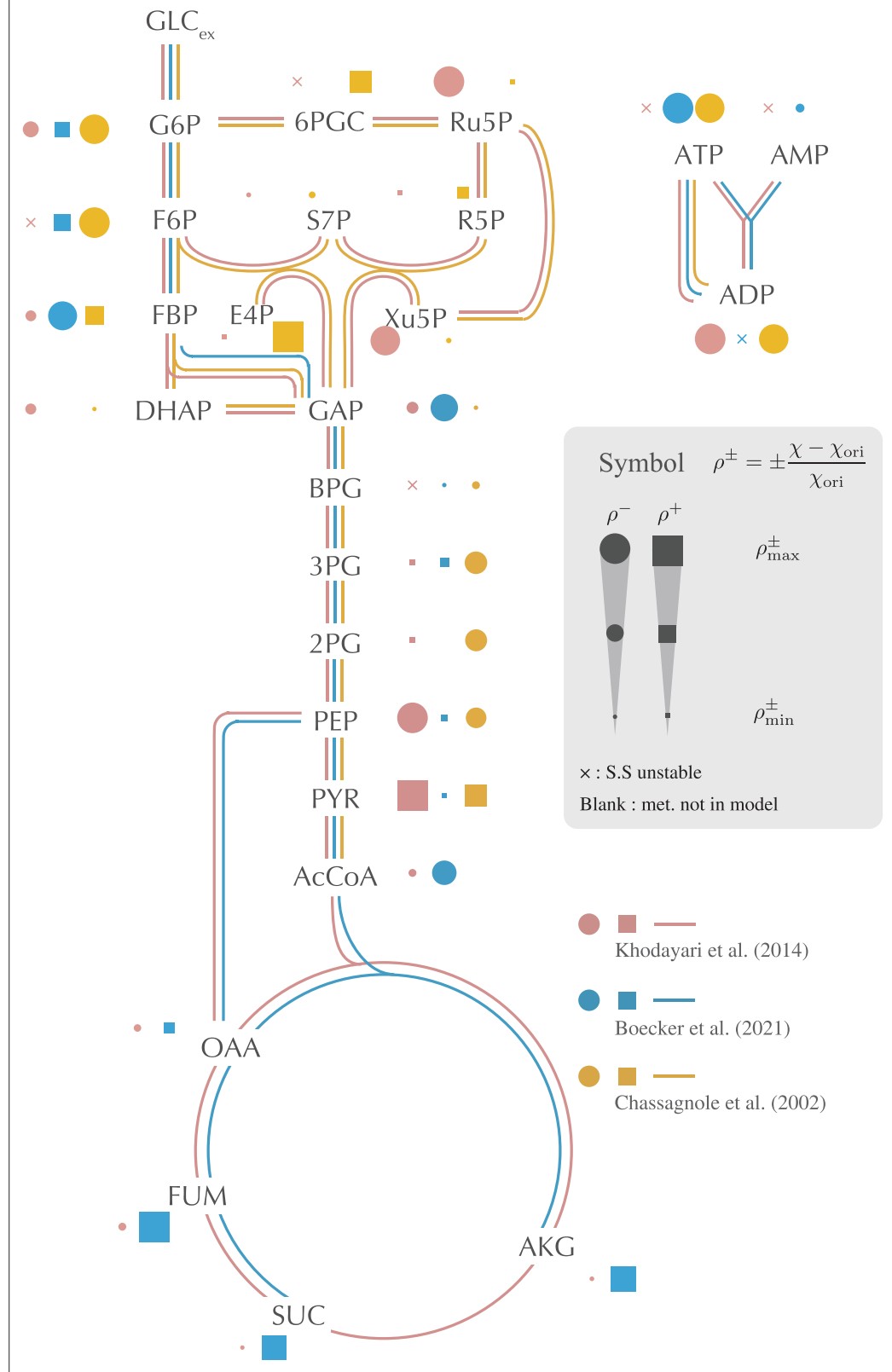

**Figure 2.** The impact of setting each metabolite's concentration to its constant, and steady-state value is depicted on the metabolic network. This impact is quantified by the relative change in the average response coefficient, as detailed in (1) and (2). The filled point and filled square symbols represent the decrease ($\rho_x^-$) and the increase ($\rho_x^+$) of the response coefficient, respectively. The size of the symbols is scaled using the maximum and minimum values

*Figure 2 continued on next page*

*Figure 2 continued*

of $\rho_x^{\pm}$ computed for each model. The cross symbol indicates that the original steady-state attractor becomes unstable when the concentration of the corresponding metabolite is fixed. If a dot, square, or cross is absent, the concentration of the corresponding metabolite is not a variable in the model. Only selected, representative metabolites are displayed in the figure. Comprehensive results are provided in *Figure 2—figure supplement 1*.

The online version of this article includes the following figure supplement(s) for figure 2:

**Figure supplement 1.** The average response coefficients of the models with the concentration of indicated metabolite fixed.

**Figure supplement 2.** The Sobol' total sensitivity index $S_{T_i}$ is computed for each metabolite.

**Figure supplement 3.** The scatter plot of the Sobol' total sensitivity index $S_{T_i}$ versus the absolute value of the change in the response coefficient.$|\rho| \equiv |(\langle \chi_x \rangle - \langle \chi_{\mathrm{ori}} \rangle|/\langle \chi_{\mathrm{ori}} \rangle)$.

**Figure supplement 4.** The response coefficient distributions of models with weakened substrate-level regulation of enzyme activity.

---

decays monotonically over time (*Awazu and Kaneko, 2009*; *Himeoka and Mitarai, 2022b*). This is in stark contrast to the behavior observed in *Figure 1B–D*.

Considering the inherent tendency of random reaction networks to display dynamics characterized by weak responses, we are compelled to investigate the influence of network sparseness on the dynamic behavior of metabolic models. To this end, we increase the reaction density of the metabolic network in the three models and assess the resultant changes in responsiveness. Our approach entails initially introducing $N_{\mathrm{add}}$ random reactions into the network and subsequently allocating kinetic parameters to these additions from the distribution of parameter values computed from the original model. We then execute a perturbation-response simulation on this extended model to determine the response coefficient, which in turn helps ascertain the distribution and average of these coefficients. This process is schematically depicted in *Figure 3A*.

*Figure 3B* illustrates the alterations in responsiveness resulting from the model extensions. We generated extended metabolic networks and accompanying ordinary differential equation (ODE) models by adding $N_{\mathrm{add}}$ random reactions, and executed the perturbation-response simulation. The average response coefficient is then computed for each model, and the distribution of the average response coefficient is presented as the violin plots. In the perturbation-response simulation, the extended models with only a single attractor are subjected to the response coefficient analysis (comprehensive procedural details available in 'Materials and methods'). Note that in this procedure metabolites are not newly introduced to the model, but only the reactions are. Thus, the reaction density increases.

As evident from the figures, a consistent trend emerges across the three models: the introduction of random reactions weakens responsiveness. However, the magnitude of this reduction varies among the models. It is important to note that the reactants of the added reactions are selected randomly, thus not based on any specific chemical rationale. Nevertheless, this diminished responsiveness persists even when the additional reactions are confined solely to those cataloged in the metabolic model database, as visualized in *Figure 3—figure supplement 1*.

Such attenuation aligns with expectations shaped by prior studies on dense reaction networks (*Furusawa and Kaneko, 2003*; *Kaneko et al., 2015*; *Awazu and Kaneko, 2009*; *Himeoka et al., 2022a*). Evidently, the inherent sparsity of the backbone reaction network plays a critical role, enabling the kinetic model of metabolic systems to respond strongly to perturbations.

## A minimal model for disrupted homeostasis

So far, we have studied the kinetic models of the central carbon metabolism of *E. coli*. Our findings indicate that cofactors and the sparsity of the backbone network (network devoid of cofactors) are the keys to the responsiveness of the metabolic dynamics. In this section, we develop a simple minimal model to investigate if these two factors are sufficient to induce strong responses in reaction systems.

Our minimal model is designed in a manner that allows us to adjust the sparsity of the backbone network and the proportion of cofactor-coupled reactions. First, we explain the construction of the backbone networks. These networks are comprised of $N$ chemical species, $C_1, C_2, \cdots, C_N$, and $R$ reactions ($R \geq N - 1$). For simplicity, we construct the backbone network using only uni-uni reactions. The

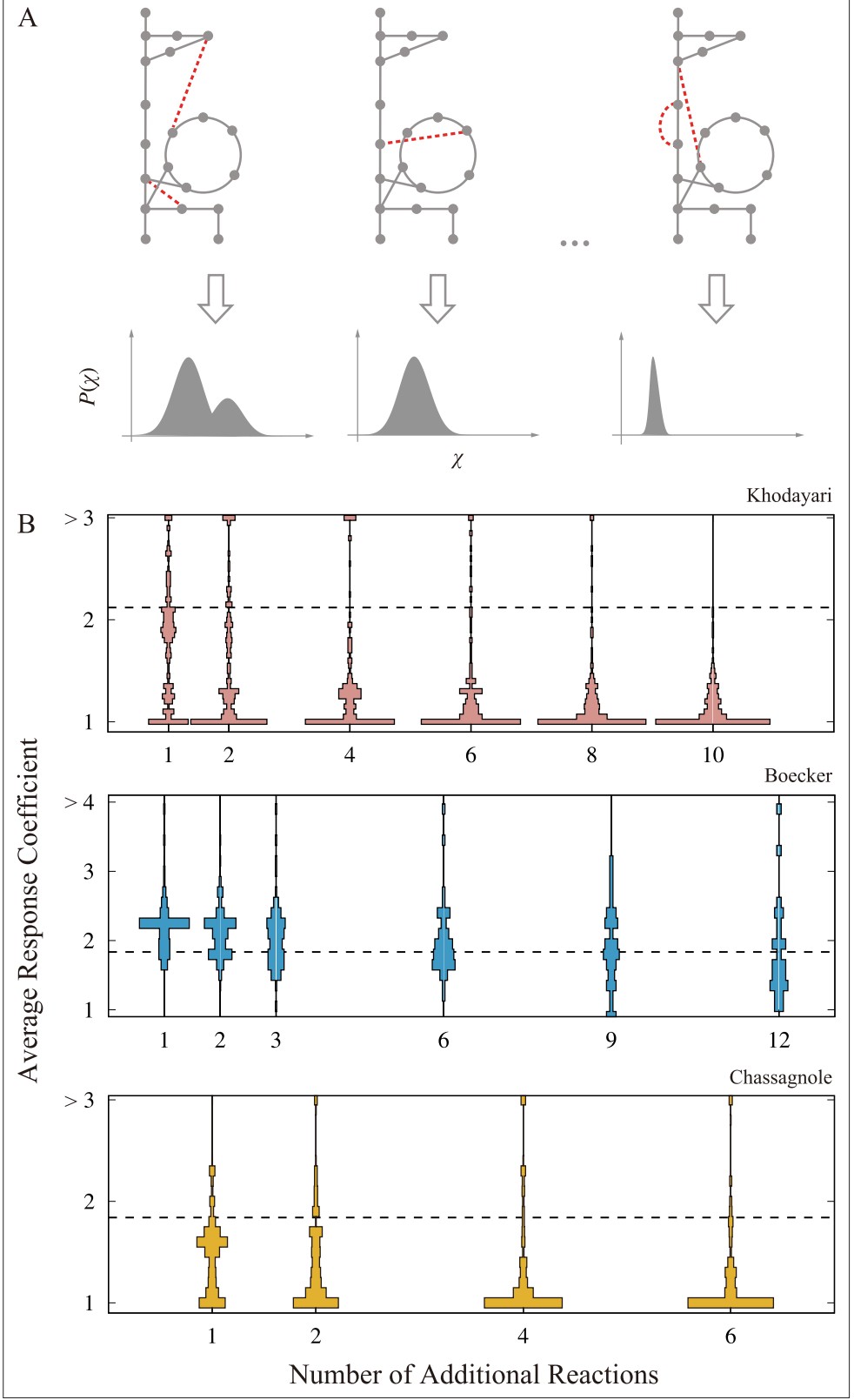

**Figure 3.** Change in the responsiveness by network expansion. (**A**) A schematic figure of the network expansion simulation. Reactions are randomly added to the original metabolic network, and then, the corresponding ordinary differential equation (ODE) model is constructed. In the illustration, the additional reaction is highlighted as the dashed red lines. The perturbation-response simulation is performed on the expanded models to obtain

*Figure 3 continued*

the response coefficient distribution. The average response coefficient is calculated as a scalar indicator of the responsiveness for each response coefficient distribution. (**B**). The distribution of the average response coefficient is plotted against the number of additional reactions for the three models. The dashed line indicates the average response coefficient of the original model with no additional reaction. For each $N_{\mathrm{add}}$, we generated 256 extended network, and 128 trajectories were computed for each extended network.

The online version of this article includes the following figure supplement(s) for figure 3:

**Figure supplement 1.** The violin plot of the average response coefficient of the network expansion of the Khodayari model.

---

minimum number of uni-uni reactions required to interlink $N$ chemical species is $N - 1$. To assemble a backbone network with $N$ chemicals and $R$ reactions, we first connect $N$ reactions using $N - 1$ reactions and subsequently introduce random reactions for the pairs of chemicals that lack reactions, totaling $R - (N - 1)$. We incorporate exchange reactions with extracellular environments for 5% of the metabolites (this percentage is not critical to the results). The exchange reactions are always introduced to the two endpoints metabolites of the sequential linkage to ensure there are no dead-ends on reaction networks. We posit that one of the endpoint metabolites from the sequential linkage is the nutrient metabolite, and we set its external concentration at 100, while for the remainder, it is set to unity.

As cofactors, we introduce the three forms of the activation for mimicking AXP, $\mathrm{A}^{**}, \mathrm{A}^{*}$, and $\mathrm{A}$, which are not included in the $N$ chemicals in the backbone network. We introduce the coupling fraction $f$ to modulate the number of reactions with cofactor coupling. With the coupling fraction $f$, the number of reactions with cofactor coupling is given as $\mathrm{Round}(fR)$ with $\mathrm{Round}(\cdot)$ as the function to round the value to the nearest integer. When a reaction is selected for cofactor coupling, two forms of the cofactor (chosen from $\mathrm{A}^{**}, \mathrm{A}^{*}$, and $\mathrm{A}$) are randomly selected. The reaction then gets coupled to the conversion between these two cofactor forms, with the direction being randomly set.

Overall, the differential equation governing the minimal model can be expressed as

$$\frac{dx_n}{dt} = \sum_{r=1}^{R} S_{nr} J_r(\boldsymbol{x}) + D_n(X_n - x_n), \tag{3}$$

In this expression, the concentrations of the chemicals from the backbone network, as well as the cofactors, are combined in a vertical stack, that is, $n = 1, 2, \cdots, N, N+1, N+2, N+3$ with $n = 1, \cdots, N$ represents the concentrations of the $n$th chemical in the backbone network, and $x_{N+1}, x_{N+2}$ and $x_{N+3}$ denote the concentrations of $\mathrm{A}^{**}, \mathrm{A}^{*}$, and $\mathrm{A}$, respectively. $S$ is the stoichiometric matrix of the reaction network. $D_n$ is unity if the $n$th chemical has an influx/efflux term, otherwise, it is zero. $X_n$ is the external concentration of chemical $n$, which is a constant parameter.

Mass action kinetics is employed for the reaction rate function $J_r$. The rate function has the form

$$J_r(\boldsymbol{x}) = v_r \left( k_r^+ \prod_{n \in \mathrm{Sub}(r)} x_n - k_r^- \prod_{n \in \mathrm{Prd}(r)} x_n \right) \tag{4}$$

where $\mathrm{Sub}(r)$ and $\mathrm{Prd}(r)$ are the set of substrates and products, respectively, of the $r$th reaction. If the reaction is coupled with the cofactor, $|\mathrm{Sub}(r)| = |\mathrm{Prd}(r)| = 2$ and otherwise $|\mathrm{Sub}(r)| = |\mathrm{Prd}(r)| = 1$. We neither allowed the self-loops in the backbone network ($\mathrm{C}_n \rightleftharpoons \mathrm{C}_n$) nor the catalytic activity of the cofactors (e.g., $\mathrm{C}_n + \mathrm{A}^{*} \rightleftharpoons \mathrm{C}_m + \mathrm{A}^{*}$).

The parameters $v_r$ and $k_r^{\pm}$ are the reaction rate and irreversibility, respectively. The $v_r$ values are randomly set as $v_r = 10^{u_r}$ with the uniformly distributed random number, $u_r \sim U(-3.66, 7.13)$. The range for the kinetic parameter is derived from the Khodayari model. $k_r^{\pm}$ is calculated using the Arrhenius equation based on the standard chemical potential of each chemical. We randomly set the standard chemical potential $\mu_n$ for each chemical, where $\mu_n$ follows the uniform distribution $U(0, 1)$, while the standard chemical potentials of $\mathrm{A}^{**}, \mathrm{A}^{*}$ and $\mathrm{A}$ are fixed to the constant values, $1.0, 0.5$, and $0.0$, respectively. To satisfy the detailed balance condition, $k_r^{\pm}$ are then given by $k_r^{\pm} = \min\{1, \exp(\mp \beta \Delta \mu_r)\}$ with $\Delta \mu_r$ as the standard chemical potential difference between the products and the substrates, $\Delta \mu_r = \sum_{n \in \mathrm{Prd}(r)} \mu_n - \sum_{n \in \mathrm{Sub}(r)} \mu_n$, and $\beta$ is the inverse temperature.

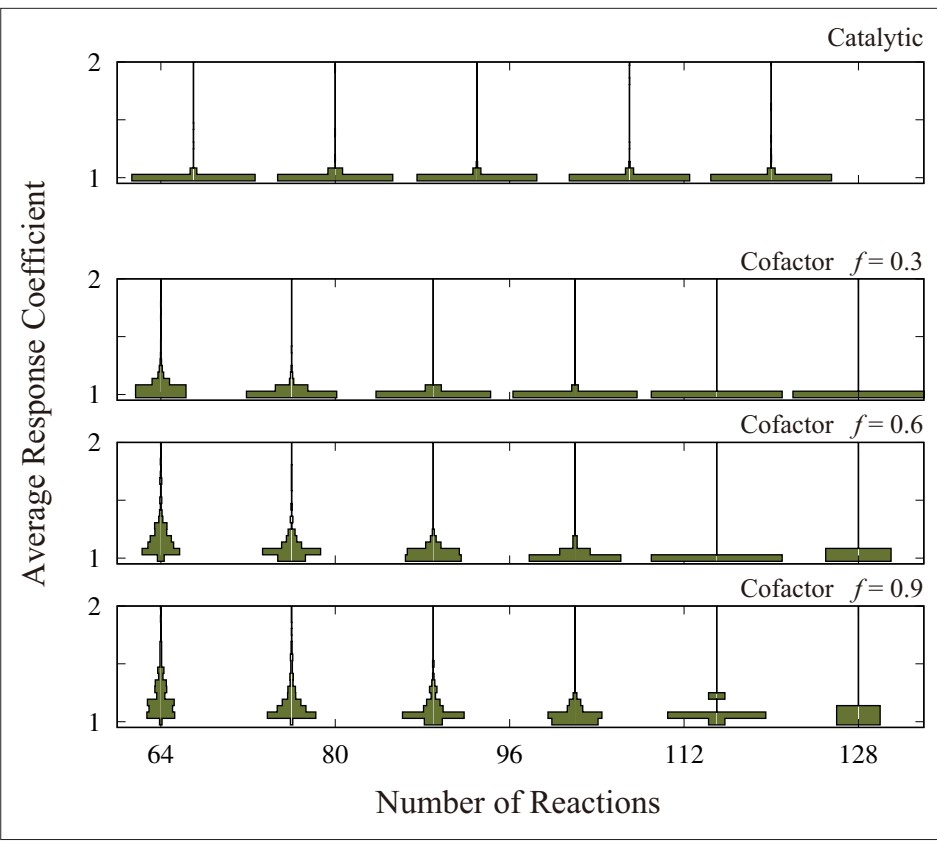

**Figure 4.** The distribution of the average response coefficient is plotted against the number of reactions, $R$, for the model with $N = 64$ metabolites plus three cofactors (the panels labeled as 'Cofactor') and $N = 67$ metabolites without cofactors (the panel labeled as 'Catalytic'). For each combination of $R$ and $f$, we generated 512 random networks and computed 128 trajectories for each of these networks. The inverse temperature $\beta$ is set at 10 for the cofactor version and 20 for the catalytic version. This is because the maximum chemical potential difference in the cofactor version is 2.0, but it is 1.0 in the catalytic version. The total concentrations of the cofactors are set to be unity.

We generate $M$ networks for several choices of reaction number $R$ and coupling fraction $f$. We performed the perturbation-response simulation to determine the response coefficient distribution of each network. In *Figure 4*, the average response coefficient, averaged over the networks, is depicted as functions of $R$ and $f$ (see the rectangles labeled as 'Cofactor'). These figures clearly illustrate that the average responsiveness tends to decrease as either $R$ increases or $f$ decreases.

The coupling with the cofactors leads to the two-body reactions, and thus, increases the nonlinearity of the model equation. It is noteworthy that the increase in responsiveness with $f$ cannot be fully attributed to the increase in the model's nonlinearity. To validate this, we employed the model (3) without the cofactors, but incorporated catalytic reactions. In this model, each reaction proceeds with the catalytic ability of an enzyme, represented as $C_n + C_l \rightleftharpoons C_m + C_l$ (here, autocatalytic reactions are permissible, meaning $l$ can be either $n$ or $m$).

In terms of the numbers of constant, linear, and quadratic terms in the model equation, the catalytic reaction version of the model corresponds to the $f = 1$ case of the original minimal model. If the increase in responsiveness with $f$ is dictated solely by nonlinearity, the catalytic reaction model should have comparable responsiveness with the 'cofactor' model with high $f$. Yet, as shown in the rectangles above the main plot in *Figure 4* (labeled as 'Catalytic'), the responsiveness metrics are notably lower than those observed in the minimal model.

## Discussion

In the present article, we have studied three models of *E. coli*'s central carbon metabolism. Our findings indicate that these models commonly exhibit strong and weak responses to perturbations of the steady-state concentration. In the weak response, the effect of the perturbation dwindles almost monotonically over time. In contrast, during a strong response, initial deviations from the steady state get amplified. Note that the linearized models cannot capture the response dynamics. We introduced the response coefficient, $\chi$, as a metric to quantify the strength of these responses and undertook comprehensive computational analyses. The strong responsiveness was observed not only in the three models studied in the article, but also in a model of glycolytic pathway with multiple sets of parameter choices (*Figure 1—figure supplement 4*; *Haiman et al., 2021*).

We aim to identify key metabolites whose dynamics play a crucial role in the emergence of strong response. We systematically fixed the concentration of each metabolite to its steady-state value and assessed the influence of individual metabolite dynamics on the overall response strength. For a majority of metabolites, the impact on the response coefficient is bidirectional: halting the dynamic concentration of a metabolite can either amplify or diminish the response, depending upon the base model. Even if their effects consistently amplify or diminish the response, the magnitude of change in the average response coefficients remains relatively modest.

However, three metabolites—ATP, ADP, and pyruvate—stood out. They consistently and significantly influenced the average response coefficient across models. Pyruvate's pronounced impact can be attributed to the PTS, whereas the effects of ATP and ADP seem to arise synergistically across multiple reactions.

One of the authors has recently reported a kinetic model of *E. coli* metabolism to exhibit anomalous relaxations and the role of AXPs in the relaxations (*Himeoka and Mitarai, 2022b*). In the paper, they reduced the metabolic network into a coarse-grained minimal network. The minimal network revealed that, depending on the initial conditions, a futile cycle conversing ATP into AMP be overly active. Then, the futile cycle can deplete ATP and ADP, leading to a significant slowdown or even cessation of several ATP/ADP-coupled reactions, pushing the entire metabolic network into a 'choked' state. The extensive connectivity of AXPs is a common trait in metabolic models. Hence, it is plausible that the impact of AXPs' dynamics on the entire metabolic system is a common feature irrespective of specific kinetic models.

Several experimental studies underscore the pivotal role of AXPs in metabolic homeostasis. For example, Teusink et al. demonstrated that in yeast a sudden increase in glucose levels in the culture medium can activate a positive feedback loop in ATP production in the glycolytic pathway, leading to growth arrest (*van Heerden et al., 2014*; *Teusink et al., 1998*). This phenomenon of growth arrest, triggered by a nutrient-level surge, has been observed across various bacterial species, including *E. coli* (*Strange and Hunter, 1966*; *Postgate and Hunter, 1964*; *Calcott and Postgate, 1972*), though its molecular mechanism has been most extensively studied in *Saccharomyces cerevisiae*. Moreover, a rapid decrease in the total adenyl cofactor concentration as a cellular response to external glucose up-shift has been documented (*Theobald et al., 1997*; *Kresnowati et al., 2006*; *Rizzi et al., 1997*; *Walther et al., 2010*). Walther et al. found that inhibiting this transient reduction in the adenyl cofactor pool increased the number of cells transitioning to a growth-arrested state upon glucose up-shift (*Walther et al., 2010*).

Next, we focused on the structure of the backbone network as suggested by the relatively simple dynamics typically observed in the random network metabolic models (*Furusawa and Kaneko, 2003*; *Kaneko et al., 2015*; *Awazu and Kaneko, 2009*; *Himeoka et al., 2022a*). To this end, we increased the density of the backbone network in the three models by randomly selecting metabolites and introducing hypothetical reactions between them. The perturbation-response simulation on the extended models indicated that as more reactions were integrated, the system's response weakened.

Finally, we devised a minimal model to evaluate our hypothesis that the dynamics of cofactors and the sparsity of the backbone network are pivotal determinants of responsiveness. This model successfully showed a strong response to perturbations when the fraction of reactions to be coupled with cofactors becomes high and the backbone network is sparse. It is worth noting that the heightened response due to cofactor coupling is not solely a result of the model equation's increased nonlinearity. Interestingly, the model's catalytic reaction variant demonstrated attenuated responses to

perturbations, even though its nonlinearity mirrored that of the minimal model where all reactions were cofactor-coupled.

The present study showed that there are similar metabolic responses to perturbations across three distinct kinetic models of *E. coli* metabolism. We like to underscore that these models were proposed by different research groups and within distinct contexts. Furthermore, the specifics of each model diverge considerably: both the Khodayari and Chassagnole models are tailored for aerobic conditions, whereas the Boecker model is for anaerobic conditions. Certain reactions and metabolites present in the Khodayari model are absent in both the Boecker and Chassagnole models. The reaction rate functions and parameter values differ across these models because each model has been developed to recapitulate different experimental results. Yet, in spite of these pronounced disparities, the models exhibit converging characteristics to a certain degree, notably the pivotal role of AXPs and the diminishing responsiveness attributed to the addition of reactions. Such congruence in outcomes might hint at the presence of features that remain resilient against variations in model specifics, potentially being shaped by the foundational network structures inherent to core metabolic processes.

From a theoretical standpoint, the robustness of metabolic systems, as well as the interplay between network structure and reaction dynamics, has been explored in the chemical reaction network theory and related fields (*Feinberg, 2019*; *Gutenkunst et al., 2007*; *Himeoka et al., 2024*; *Kobayashi et al., 2022*). However, the practical relevance of these theories to metabolic systems remains to be ascertained since the theories are mathematically rigorous and the strong assumptions narrow down the applicable systems. The cross-talk between bacterial physiology and chemical reaction network theory should enhance the developments of each field.

In the following sections, we discuss the potential implications of strong responsiveness in the context of biological sciences. The unique feature of strong responsiveness, as observed in our realistic metabolic model, naturally prompts the question: what could be its biological function? An initial hypothesis suggests its role in sensing ambient condition changes and fluctuations. Well-known design principles of signal transduction systems emphasize the amplification of external stimuli (*Ferrell and Ha, 2014*). Given that certain transcription factors are known to sense concentrations of metabolites in central carbon metabolism (*Kotte et al., 2010*; *Okano et al., 2020*; *Kochanowski et al., 2013*), strong responsiveness could serve as an amplifier for external cues.

Exploring metabolic dynamics also holds promise for synthetic biology and metabolic engineering (*Shimizu et al., 2001*; *Schwander et al., 2016*; *Bar-Even et al., 2010*; *Doerr et al., 2019*; *Claassens et al., 2019*; *Kurisu et al., 2023*; *Opgenorth et al., 2014*; *Opgenorth et al., 2017*). The construction of artificial metabolic reaction networks often relies on pathway prediction algorithms, constraint-based models, and machine learning (*Moriya et al., 2010*; *Araki et al., 2015*; *Orth et al., 2010*; *Varma and Palsson, 1993*; *Segler et al., 2018*). While these algorithms consider stoichiometric feasibility, they frequently overlook dynamics. This oversight can result in proposed network structures underperforming *Schwander et al., 2016*; *Bar-Even et al., 2010*; *Doerr et al., 2019*; *Claassens et al., 2019*, especially in terms of cofactor recycling (*Opgenorth et al., 2014*; *Opgenorth et al., 2017*). Notably, in silico reconstructed metabolic systems often disrupt cofactor balance (*Okamoto et al., 1980*; *Okamoto and Hayashi, 1983*), posing challenges to bottom-up synthesis in artificial cells. We hope our study offers foundational insights for the development of quantitative metabolic dynamics theories.

## Materials and methods
### Models and modifications

In *Khodayari et al., 2014*, the kinetic parameter values are estimated by using the ensemble modeling (*Khodayari et al., 2014*; *Tran et al., 2008*; *Tan and Liao, 2012*; *Rizk and Liao, 2009*) so that the steady flux distribution of the metabolic reactions becomes consistent with the experimental measurements. However, the steady-state dealt with in the article turned out to be unstable, that is, the maximum eigenvalue of the Jacobi matrix of the linearized system at the steady state was positive. For carrying out the perturbation-response simulation, we need to make the steady-state stable. To make the steady-state stable, we fixed the periplasmic hydrogen concentration to the steady-state value and removed the competitive inhibition of phosphoenolpyruvate carboxylase (those two factors were manually found). Also, we set the concentrations of the metabolites in the culture media (extracellular

metabolites) with the assumption that the model cell is growing in a sufficiently large flask. In addition, the small molecules, $H_2O$, $O_2$, $CO_2$, $NH_4$, and phosphate, are set to be constant, further assuming that the exchange rate of the small chemicals between the extracellular culture is sufficiently fast.

For the dynamics of ATP and ADP concentrations, in the Chassagnole model (*Chassagnole et al., 2002*), we used the reactions that coupled with ATP and ADP. Since the kinetic rate equations of those reactions are defined with the dependency of ATP and ADP concentrations in the original model, we utilized the kinetic rate equations as they are. The reactions for the adenine base synthesis are not modeled in the model, and thus, the total concentration of ATP and ADP is the conserved quantity in the model. By assuming that the decreases of the ATP and ADP due to the growth dilution are compensated, the growth dilution term is not introduced to the model equations of ATP and ADP concentrations. Lastly, we introduced the non-growth-associated ATP hydrolysis reaction (ATP→ADP) since the model fails to balance ATP and ADP and relaxes to a steady state where the concentration of some metabolites is almost zero. We utilized a simple Michaelis–Menten rate equation for the hydrolysis reaction $J_{AH} = v x_{ATP}/(K_M + x_{ATP})$, where $v = 0.1$ (sec$^{-1}$) and $K_M = 1.0$ (mM)

The Boecker model (*Boecker et al., 2021*) is used without modification.

## Computation of dynamics

All the ODE simulations in the present study are done using ode15s() function in MATLAB (MathWorks, *Inc, 2022*). For computing the attractor of the Chassagnole model, we used the initial concentrations registered in chassagnole4 on JWS online (*Olivier and Snoep, 2004*) as the initial concentrations of the modified Chassagnole model to run the dynamics. The steady-state concentrations of the metabolites reach the initial concentrations and are utilized as the attractor.

For generating random initial conditions by perturbations without altering the conserved quantities, we have performed the conservation analysis described in *Vallabhajosyula et al., 2006*.

## Estimate of the fluctuation strength

For estimating the strength of fluctuations, we utilize the following simple metabolic reaction:

$$\emptyset \xrightarrow{P} X \xrightarrow{Q} \emptyset.$$

The concentration of the enzyme $P$ and $Q$ follows the stochastic transcription and translation model (*Friedman et al., 2006*; *Paulsson and Ehrenberg, 2000*). The temporal change of the concentration of the metabolite $X$ here is given by a simple model with the mass-action kinetics,

$$\frac{dx}{dt} = vp(t) - uq(t)x(t), \tag{5}$$

where $x, p, q$ are the concentrations of $X, P$, and $Q$, respectively. $v$ and $u$ are the kinetic parameters of the reactions. According to the time-scale separation of metabolic reactions and transcription/translation, we suppose that $x$ is the fast variable and its dynamics slave to $p$ and $q$. We assume that the timescales in the change of the metabolite concentrations are sufficiently smaller than that of the proteins and apply the quasi-steady-state approximation: we solve the steady state of the deterministic (*Equation 5*; *Kaneko, 1981*; *Risken, 1996*). Then, we calculate the mean and variance of the solution $x(t) = vp(t)/uq(t)$, which is a stochastic variable. Since the stochastic transcription and translation model leads to the gamma distribution of the protein concentration, the average and the variance are given by

$$\langle x \rangle = \frac{v/u}{b^{2a}\Gamma^2(a)} \int_{\mathbb{R}^2_+} dp\, dq\, \frac{p}{q} p^{a-1} e^{-p/b} q^{a-1} e^{-q/b}$$

$$= \frac{v}{u}\frac{a}{a-1} \tag{6}$$

$$\langle x^2 \rangle = \frac{(v^2/u^2)}{b^{2a}\Gamma^2(a)} \int_{\mathbb{R}^2_+} dp\, dq\, \frac{p^2}{q^2} p^{a-1} e^{-p/b} q^{a-1} e^{-q/b}$$

$$= \frac{v^2}{u^2}\frac{a(a+1)}{(a-1)(a-2)} \tag{7}$$

where $a$ and $b$ are the parameters set by the rate of the transcription, translation, mRNA degradation, and protein degradation.

The coefficient of variation is then given by

$$\frac{\sqrt{\langle x^2 \rangle - \langle x \rangle^2}}{\langle x \rangle} = \sqrt{\frac{2a-1}{a(a-2)}}. \tag{8}$$

where $a$ is the ratio between the transcription rate and the protein degradation rate. According to the large-scale proteomic analysis of *E. coli* (***Taniguchi et al., 2010***), the mean value of $a$ among the essential genes is $\approx 6.82 \pm 2.34$ (only with the essential metabolic enzymes listed in the iML1515 [***Monk et al., 2017***], the average value is $\approx 6.72 \pm 2.34$). By substituting $a = 6.82$ into ***Equation 8***, we obtain the relative noise strength as $\approx 62\%$. Since in the estimate we suppose the irreversible reactions lead to larger noise amplitudes than reversible reactions, we regard 62% as the maximum relative noise strength and use a bit smaller value, 40%.

## Random catalytic reaction network model

The RCRN model is a simple, mass-balancing toy model of cellular metabolism (***Furusawa and Kaneko, 2003***; ***Kaneko et al., 2015***). The ordinary differential equation dictating the temporal evolution of the $i$th metabolite's concentration $x_i$ is given by

$$\frac{dx_i}{dt} = \sum_{j,k} x_k \left( v_{ijk} x_j - v_{jik} x_i \right) + D_i x_{T(i)} x_i^{(\text{ext})} - \mu x_i$$

where $v_{ijk}, D_i$, and $\mu$ are the rate constant of the reaction $C_j \to C_i$ catalyzed by $C_k$ ($C_i$ denotes the $i$th metabolite), the substrate update rate of $C_i$, and the specific growth rate of the model cell, respectively. $T(i)$ denotes the transporter metabolite (enzyme) for the uptake of $C_i$, and $x_i^{(\text{ext})}$ is the external concentration of $C_i$. In the RCRN model, the total volume of the cell is usually set to be equal to the total amount of the metabolites. This is because each 'metabolite' in the model is interpreted as an enzyme as well as a reactant of the metabolic reactions. Hence, the specific growth rate of the model cell is set to the total uptake rate of the metabolites, $\mu = \sum_i D_i x_{T(i)} x_i^{(\text{ext})}$.

In the construction of an instance of the RCRN model, we first generate a random, connected network among $N$ metabolites, and then, we assign a single catalyst on each reaction edge. If the reaction $C_j \to C_i$ catalyzed by $C_k$ exists in the network, $v_{ijk}$ is non-zero, while otherwise, it is set to zero. In the present article, we suppose all the reactions are reversible, and thus, $v_{jik}$ is non-zero if and only if $v_{ijk}$ is non-zero.

For making the comparison of the responsiveness of the realistic model to the RCRN model, we generated 128 instances of the RCRN model where the number of metabolites $N$ is set to be equal to the realistic model that we want to make a comparison. The number of reactions $R$ is set to $3.5N$ for the Boecker and Chassagnole model while $4N$ for the Khodayari model. The ratio between $N$ and $R$ is set so that the Erdős–Rényi random graph of given size becomes connected with sufficiently high probability. The reaction rate constants $v_{ijk}$'s are sampled from the model to compare. For the sake of simplicity, we suppose only the first metabolite is the nutrient metabolite which is taken up from the external environment and the $N$th metabolite is the transporter of it. $D_i$ and $x_i^{(\text{ext})}$ are set to unity for $i = 1$, and for the else, those parameters are set to zero.

For each generated instance of the RCRN model, we performed the perturbation-response simulation where the attractor(s) are computed by simulating the model dynamics from randomly generated 32 initial concentrations $\boldsymbol{x}_{\text{ini}} \in (10^{-3}, 10^3)^N$. The RCRN models exhibiting multistability or computational failure due to numerical issues such as heavy stiffness are not subjected to further analysis (less than 10% of the model instances showed multistability for corresponding models; the computational failure is $\approx 20\%$ for the RCRN model with the parameter distribution of Chassagnole model, $\approx 50\%$ for that of the Khodayari model, while less than 1% for that of the Boecker model). For quantifying the response coefficient, we computed 128 trajectories for each of those models.

## Random addition of the metabolic reactions

First, we add uni-uni reactions where the reactants are randomly picked from the non-cofactor metabolites. Next, we decide whether the additional reaction to be coupled with the adenyl cofactor metabolites (ATP, ADP, or AMP) with a probability $p$. The probability $p$ equals to the fraction of adenyl cofactor-coupled reactions in the original model. The addition reaction is either the uni-uni reaction (if the reaction is not coupled with AXPs) or the bi-bi reaction (if the reaction is coupled with the adenyl cofactors). We computed the parameter value distribution for each reaction scheme, and the parameter values for the additional reactions are sampled from the corresponding parameter value distribution.

For the Khodayari model, the additional reaction is modeled using the elementary reaction decomposition (ERD) (see the next section), that is, a reaction $A \rightleftharpoons B$ is decomposed into the three elementary reaction steps, $E + A \rightleftharpoons EA$, $EA \rightleftharpoons EB$, and $EB \rightleftharpoons E + B$. The mass-action kinetics is adopted as the reaction rate function for each elementary reaction step.

The ERD is not applied to the Boecker model and the Chassagnole model. Also, several types of reaction rate functions are used in the two models. Thus, we decided to use the simplest enzymatic reaction kinetics used in each paper. The reversible Michaelis–Menten kinetics

$$v = \frac{v_+[S_1] - v_-[P_1]}{1 + [S_1]/K_{S_1} + [P_1]/K_{P_1}}$$

is used for uni-uni reactions, and for bi-bi reactions, the ordered bi-bi reaction kinetics

$$v = \frac{v_+[S_1][S_2] - v_-[P_1][P_2]}{1 + \frac{[S_1]}{K_{S_1}} + \frac{[P_1]}{K_{P_1}} + \frac{[S_1][S_2]}{K_{S_1}K_{S_2}} + \frac{[P_1][P_2]}{K_{P_1}K_{P_2}}}$$

is adopted in the Boecker model, and a reaction kinetics

$$v = \frac{v_+[S_1][S_2] - v_-[P_1][P_2]}{(1 + [S_1]/K_{S_1} + [P_1]/K_{P_1})(1 + [S_2]/K_{S_2} + [P_2]/K_{P_2})}$$

is used in the Chassagnole model. $[S_i], [P_i]$, and $K_*$ ($*$ is either $S_1, S_2, P_1$ or $P_2$) represent the concentration of the $i$th substrate and product, and the dissociation constant of the corresponding chemical, respectively.

After constructing an extended model, we compute the steady-state attractor by simulating the dynamics from 128 random initial concentrations. These initial conditions are generated by applying a 40% relative perturbation to the steady-state concentrations of the original model. If all the initial conditions converge to a single attractor, then we generate another set of 128 initial conditions by applying a 40% relative perturbation to the steady-state concentrations of the extended model in order to perform the perturbation-response simulation. For the Boecker model, we require the extended model to have a steady growth rate greater than half of that of the original model. This is because growth dilution is typically the slowest process in the model, and thus, a significant slowdown of growth dilution can lead to an artificial overestimation of the model's responsiveness.

## Elementary reaction decomposition

Here, a brief description of the ERD is provided. Suppose that we have the following bi-to-uni reaction

$$A + B \rightleftharpoons C, \tag{9}$$

and that the reaction is catalyzed by an enzyme E. For implementing this reaction into the kinetic model, the Michaelis–Menten-type kinetics of chemical reaction is often adopted. With the Michaelis–Menten-type kinetics, the rate of the reaction $J$ is given by

$$J = [E] \frac{v_+[A][B] - v_-[C]}{1 + [A]/K_A + [A][B]/K_{AB} + [C]/K_C}, \tag{10}$$

where we supposed that the binding of the substrate to the enzyme occurs in the order of $A \rightarrow B$.

The Michaelis–Menten-type reaction kinetics (*Equation 11*) is obtained by the pseudo-equilibrium approximation or the steady-state approximation of the following reactions:

$$A + E \rightleftharpoons EA \qquad (11)$$
$$EA + B \rightleftharpoons EAB \qquad (12)$$
$$EAB \rightleftharpoons EC \qquad (13)$$
$$EC \rightleftharpoons E + C \qquad (14)$$

In the ERD scheme, each step (*Equations 12–14*) of the overall reaction (*Equation 9*) is modeled using the mass-action kinetics. Also, the forward and the backward reactions are dealt as different reactions. For instance, the rates of the reaction $A + E \rightleftharpoons EA$ are given by $J_{A,E \to EA} = v_{A,E \to EA}[E][A]$ and $J_{EA \to A,E} = v_{EA \to A,E}[EA]$.

The advantage of the ERD is that every reaction rate function is given either by a linear or quadratic function of the metabolites' concentrations. This feature allows us in the present article to randomize the reaction network in a unified manner. Also, the ERD has the advantage of reducing the computational cost for the parameter estimation based on omics data. For more details, see original papers (*Khodayari et al., 2014*; *Tran et al., 2008*; *Tan and Liao, 2012*; *Rizk and Liao, 2009*).

## Generative AIs and other software

ChatGPT (OpenAI, September 25, 2023, version) and Grammarly (Grammarly, Inc) are used to improve clarity and brush up on English grammar.

## Acknowledgements

The authors thank Namiko Mitarai and Taro Furubayashi for fruitful discussions. This work is supported by JSPS KAKENHI Grant Numbers JP 21K20626, JP 22K15069, and JP 22H05403 to YH; JP 22K21344 to CF; the Japan Science and Technology Agency (JST) ERATO (JPMJER1902 to CF), and GteX Program (JPMJGX23B4 to YH).

## Additional information

### Funding

| Funder | Grant reference number | Author |
| --- | --- | --- |
| Japan Society for the Promotion of Science | JP 21K20626 | Yusuke Himeoka |
| Japan Society for the Promotion of Science | JP 22K15069 | Yusuke Himeoka |
| Japan Society for the Promotion of Science | JP 22H05403 | Yusuke Himeoka |
| Japan Society for the Promotion of Science | JP 22K21344 | Chikara Furusawa |
| Japan Science and Technology Agency | 10.52926/JPMJER1902 | Chikara Furusawa |
| Japan Science and Technology Agency | 10.52926/JPMJGX23B4 | Yusuke Himeoka |

The funders had no role in study design, data collection and interpretation, or the decision to submit the work for publication.

### Author contributions

Yusuke Himeoka, Conceptualization, Resources, Data curation, Software, Formal analysis, Funding acquisition, Validation, Investigation, Visualization, Methodology, Writing – original draft, Project administration; Chikara Furusawa, Conceptualization, Project administration, Writing - review and editing

### Author ORCIDs

Yusuke Himeoka ⓘ https://orcid.org/0000-0001-8545-1625

Reviewer #1 (Public review): https://doi.org/10.7554/eLife.98800.4.sa1
Reviewer #2 (Public review): https://doi.org/10.7554/eLife.98800.4.sa2
Author response https://doi.org/10.7554/eLife.98800.4.sa3

## Additional files

### Supplementary files
MDAR checklist

Supplementary file 1. The list of substrate-level regulations in each model and reactions used for the non-random network expansion.

### Data availability
All the codes for the analysis in the paper are available on https://github.com/yhimeoka/Perturbation-Response-Analysis (copy archived at *Himeoka, 2025*).

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
