## [Editor Report · eLife Assessment]

This **valuable** study uses dynamic metabolic models to compare perturbation responses in a bacterial system, analyzing whether they return to their steady state or amplify beyond the initial perturbation. The evidence supporting the emergent properties of perturbed metabolic systems to network topology and sensitivity to specific metabolites is **solid**.

---

## [Referee Report · Reviewer #1 (Public review)]

Summary:

The author studied metabolic networks for central metabolism, focusing on how system trajectories returned to their steady state. To quantify the response, systematic perturbation was performed in simulation and the maximal destabilization away from steady state (compared with initial perturbation distance) was characterized. The author analyzed the perturbation response and found that sparse network and networks with more cofactors are more "stable", in the sense that the perturbed trajectories have smaller deviation along the path back to the steady state.

Strengths and major contributions:

The author compared three metabolic models and performed systematic perturbation analysis in simulation. This is the first work characterized how perturbed trajectories deviate from equilibrium in large biochemical systems and illustrated interesting findings about the difference between sparse biological systems and randomly simulated reaction networks.

Discussion and impact for the field:

Metabolic perturbation is an important topic in cell biology and has important clinical implication in pharmacodynamics. The computational analysis in this study provides an initiative for future quantitative analysis on metabolism and homeostasis.

Comments on latest version:

In the latest version of this work, the author included NADH, NADPH into the analysis, and perform some comparison about sensitivity analysis. I think this paper is ready to be finalized, and many open questions inspired from this work can be studied in future.

---

## [Referee Report · Reviewer #2 (Public review)]

The authors have conducted a valuable comparative analysis of perturbation responses in three nonlinear kinetic models of *E. coli* central carbon metabolism found in the literature. They aimed to uncover commonalities and emergent properties in the perturbation responses of bacterial metabolism. They discovered that perturbations in the initial concentrations of specific metabolites, such as adenylate cofactors and pyruvate, significantly affect the maximal deviation of the responses from steady-state values. Furthermore, they explored whether the network connectivity (sparse versus dense connections) influences these perturbation responses. The manuscript is reasonably well written.

Comments on latest version:

The authors have adequately addressed my concerns.

---

## [Author Response]

The following is the authors’ response to the previous reviews

Reply to the comments of the second referee

We sincerely appreciate the positive evaluation and the useful suggestions on our manuscript.

(1) The authors identified key metabolites affecting responses to perturbations in two ways: (i) by fixing a metabolite's value and (ii) by performing a sensitivity analysis. It would be helpful for the modeling community to understand better the differences and similarities in the obtained results. Do both methods identify substrate-level regulators? Is freezing a metabolite's dynamics dramatically changing the metabolic response (and if yes, which ones are so different in the two cases)? Does the scope of the network affect these differences and similarities?

Thank you for these suggestions. We compared the Sobolʼ total sensitivity index with the absolute values of the change in the response coefficient (Figure S6 in the revised manuscript). There is no clear relationship between the two quantities. The Sobolʼ sensitivity analysis quantifies how a perturbation on the concentration of a metabolite X contributes to the overall dynamics. On the other hand, the analysis in which metabolitesʼ concentrations are fixed measures how strongly metabolite X helps propagate the perturbations on the other metabolites throughout the metabolic network. In other words, in the Sobolʼ analysis, we evaluate the outcome when the perturbation is applied directly to metabolite X, whereas in the fixing-metabolites analysis, we consider perturbations applied to other metabolites and assess how X influences those perturbations. We believe this conceptual difference explains why the two quantities do not correlate. We suspect that this lack of correlation is independent of the networkʼs scope, because each method evaluates a different aspect of the system. We would say that both methods identify the effect of the metabolite dynamics on the overall dynamics whatever the form is, i.e. the methods do not distinguish the perturbation on the metabolite affecting the overall dynamics by whether the stoichiometric (reactant) way or, the substrate-level regulations. Thus, identifying the substrate-level regulation by utilizing the methods would be challenging.

(2) Regarding the issues the authors encountered when performing the sensitivity analysis, they can be approached in two ways. First, the authors can check the methods for computing conserved moieties nicely explained by Sauro's group (doi:10.1093/bioinformatics/bti800) and compute them for large-scale networks (but beware of metabolites that belong to several conserved pools). Otherwise, the conserved pools of metabolites can be considered as variables in the sensitivity analysis-grouping multiple parameters is a common approach in sensitivity analysis.

Thank you for this helpful suggestion. Following the method described in the reference, we have computed the Sobolʼ sensitivity index of NADH, NADPH, and Q8H2 (with their counterparts algebraically solved and treated as dependent variables). We have updated Figure S5 accordingly.